# Deoxygenative photochemical alkylation of secondary amides enables a streamlined synthesis of substituted amines

Antonio Pulcinella[1,5], Stefano Bonciolini ®[1,5], Robin Stuhr ®[1,2], Damiano Diprima[1], Minh Thao Tran ®[3], Magnus Johansson ®[4], Axel Jacobi von Wangelin[2] & Timothy Noël ®[1] ✉

Secondary amines are vital functional groups in pharmaceuticals, agrochemicals, and natural products, necessitating efficient synthetic methods. Traditional approaches, including *N*-monoalkylation and reductive amination, suffer from limitations such as poor chemoselectivity and complexity. Herein, we present a streamlined deoxygenative photochemical alkylation of secondary amides, enabling the efficient synthesis of α-branched secondary amines. Our method leverages triflic anhydride-mediated semi-reduction of amides to imines, followed by a photochemical radical alkylation step. This approach broadens the synthetic utility of amides, facilitating late-stage modifications of drug-like molecules and the synthesis of saturated *N*-substituted heterocycles. The pivotal role of flow technology in developing a scalable and robust process underscores the practicality of this method, significantly expanding the organic chemist's toolbox for complex amine synthesis.

Secondary amines are among the most commonly encountered functional groups in approved drugs, agrochemicals, and natural products (Fig. 1A)[1,2]. Modulating the nucleophilicity and basicity of the nitrogen atom in a lead compound is often employed in drug discovery campaigns to optimize and fine-tune the associated physicochemical properties[3]. Furthermore, secondary amines serve as versatile building blocks for synthesizing other medicinally relevant nitrogen-containing compounds, such as amides, sulfonamides, and *N*-heterocycles[4,5].

Given their importance, the preparation of a diverse array of secondary amines is a cornerstone in organic chemistry, and there is a growing need for new robust synthetic methodologies[6–10]. Traditional synthetic approaches typically rely on the *N*-monoalkylation of primary amines with (pseudo)alkyl halides or proceed via the reductive amination of carbonyl compounds (Fig. 1B)[11,12]. Despite their operational simplicity, these methods suffer from significant limitations, such as poor chemoselectivity (leading to *N*-over-alkylation) and a lack of versatility in synthesizing complex

substituted amines. Specifically, the construction of α-substituted secondary amines via ketone reductive amination can be hindered by sluggish imine formation due to stereoelectronic factors[11,13]. Additionally, the multistep synthesis required to access the corresponding decorated ketone further diminishes the synthetic appeal of this strategy.

An elegant alternative involves Mannich-type nucleophilic addition to aldehyde-derived iminium ions (Fig. 1B)[14–16]. This multicomponent reaction facilitates the installation of the branched nitrogen unit through the coupling of aldehydes, aliphatic amines, and accessible nucleophilic precursors. Although advancements in organometallic multicomponent Mannich reactions have been made[17–21], the corresponding radical approach offers higher functional group tolerance and complementary reactivity. However, this paradigm largely remains limited to imines bearing auxiliary groups, ultimately yielding *N*-protected secondary amines[22–25]. Recently, protocols have been developed to expand the utility and scope by leveraging radical

[1]Flow Chemistry Group, Van't Hoff Institute for Molecular Sciences (HIMS), University of Amsterdam, Amsterdam, The Netherlands. [2]Department of Chemistry, University of Hamburg, Hamburg, Germany. [3]Janssen Pharmaceutica NV, Beerse, Belgium. [4]Medicinal Chemistry, Research and Early Development, Cardiovascular, Renal and Metabolism (CVRM), BioPharmaceuticals R&D, AstraZeneca, Gothenburg, Sweden. [5]These authors contributed equally: Antonio Pulcinella, Stefano Bonciolini. ✉e-mail: t.noel@uva.nl

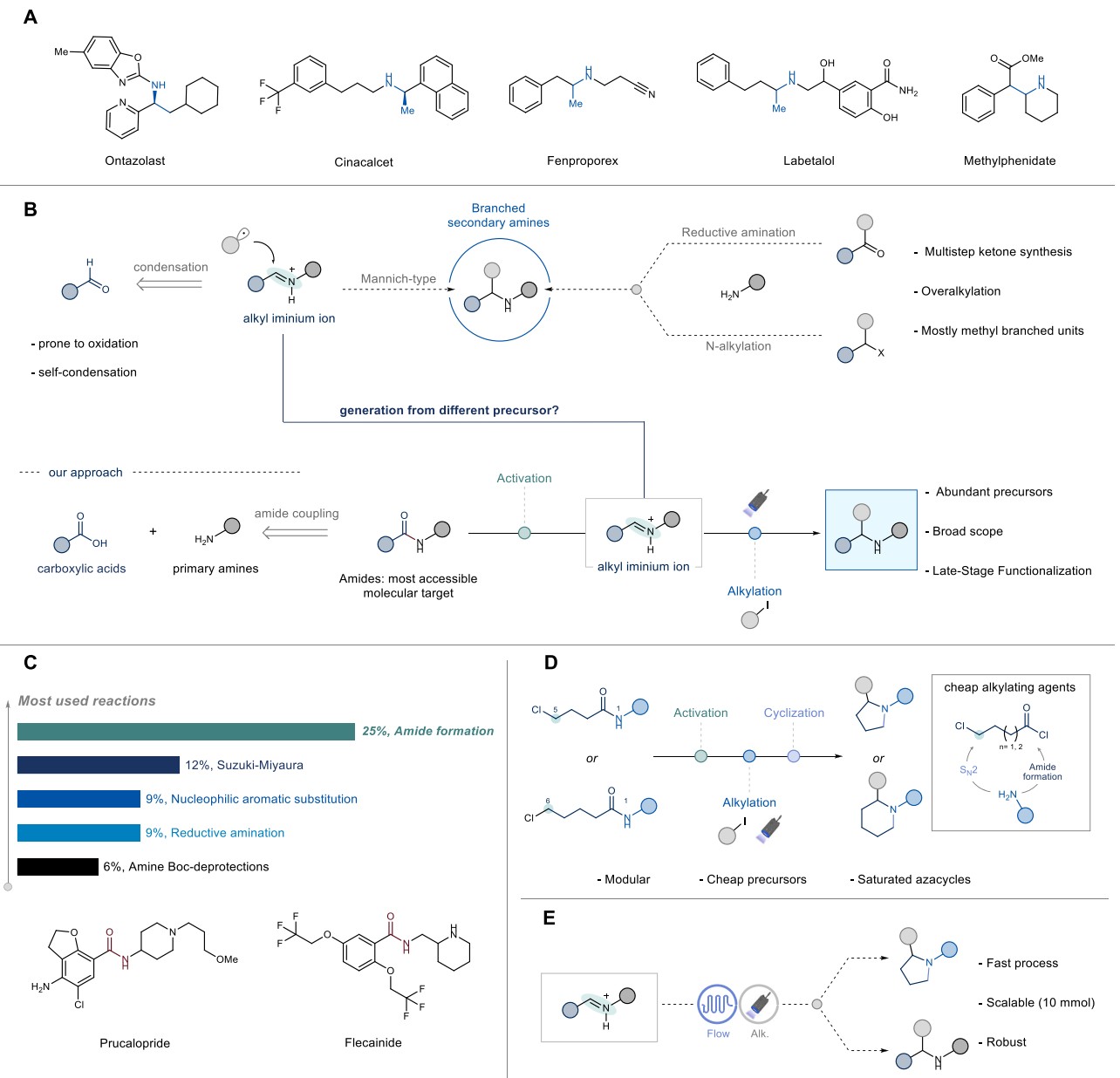

**Fig. 1 | Design and applications of the deoxygenative photochemical alkylation of sec-amides. A** Drugs containing branched secondary amine units. **B** Proposed approach: sec-amides as convenient precursor to branched amines. **C** Amides in drug discovery. **D** Alkylative annulation strategy to α-substituted N-heterocycles. **E** Flow chemistry enables fast, scalable processes.

addition to unbiased imines or *N*-alkyl iminium ions under mild photochemical conditions[13,26–28]. While these methods effectively produce a wide range of substituted tertiary amines, only a few examples have proven effective for accessing substituted secondary amines[16,27,28].

Despite these considerable breakthroughs, all the aforementioned synthetic efforts build on the mechanistic foundations of the parent carbonyl reductive amination. The requirements of the sensitive aldehyde coupling partner constitutes the major limitation due to the well-known propensity of this functional group to undergo degradative oxidation and self-condensation[29,30]. Additionally, from a practical standpoint, the preparation of this functional group typically necessitates redox manipulation of the corresponding alcohol or carboxylic acid. To overcome these drawbacks, the development of a platform for the streamlined formation of α-branched secondary amines from more ubiquitous precursors like carboxylic acids would significantly expand the organic chemist's toolbox (Fig. 1B).

Guided by this principle, we selected amides as ideal candidates, as their synthesis is one of the most common reactions performed in modern pharmaceutical endeavors (Fig. 1C)[31–35]. Considering the extensive libraries of carboxylic acids and amines, we envision that repurposing the typically inert amide functionality will allow investigators to explore new chemical spaces during drug discovery campaigns[30,31,36–45]. Given our recent interest in photochemical deoxygenative transformations[46,47], we set out to target the nucleophilic oxygen of amides to achieve chemoselective functionalization of secondary amides in the presence of more activated electrophilic functional groups.

Specifically, we propose a two-phase strategy using abundant secondary amides: a first deoxygenative event to access *N*-alkyl iminium ions, followed by a photochemical radical alkylation to yield the desired α-branched secondary amine (Fig. 1B). Mechanistically, based on seminal reports from Charette and coworkers[48–50], we investigated

**Table 1 | Optimization of the one-pot deoxygenative alkylation of sec-amide 1a**

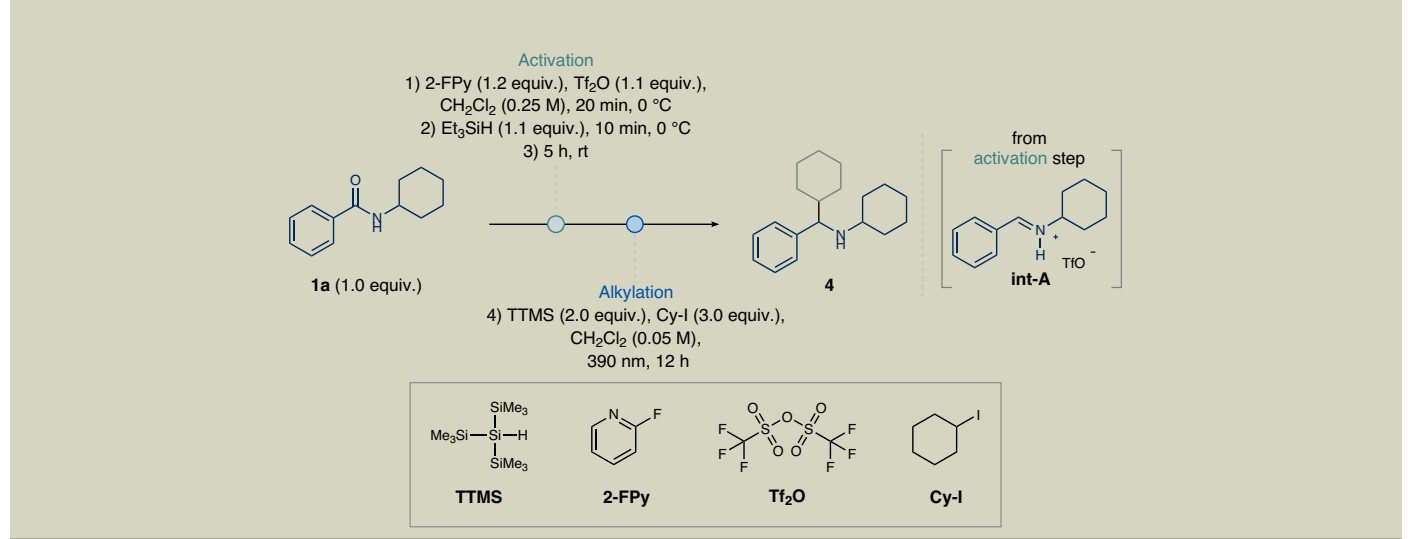

| Entry | Deviation | Yield of 3[a] |
|---|---|---|
| 1 | None | 75% |
| 2 | 2 h, rt | 76%[b] |
| 3 | 456 nm | 65% |
| 4 | AIBN instead of light | 44% |
| 5 | *i*PrOH (5 equiv.) as additive | 65%[c] |
| **6** | **CH₂Cl₂:CH₃CN 1:4** | **80%[c]** |
| 7 | CH₂Cl₂:EtOAc 1:4 | 32%[c] |
| 8 | Dark | n.d.[d] |
| 9 | Without TTMS | n.d.[d] |

Bold highlights the best optimized conditions, then used for the scope.
[a]Yields were determined by [1]H NMR using trichloroethylene as external standard (0.25 mmol scale).
[b]Refers to the step 3.
[c]Refers to the photochemical step 4.
[d]Quantitative recovery of imine (see SI). See Supplementary Information for experimental details.

whether triflic anhydride-mediated semi-reduction of amides to imines could be combined with a photochemical carbon radical addition facilitated by silane-mediated halogen atom transfer (XAT)[13,51,52].

In this study, we report the successful realization of this design, enabling the conversion of a broad range of amides and the late-stage modification of drug-like molecules into complex α-branched secondary amines (Fig. 1B). Our method also offers a facile reaction sequence for the synthesis of *N*-substituted saturated heterocycles through the direct annulation of inexpensive bis-electrophiles and primary amines (Fig. 1D)[53–55]. Additionally, we demonstrate the pivotal role of flow technology in developing a scalable, fast, and robust process for the streamlined synthesis of alkyl amines (Fig. 1E).

## Results

### Reaction optimization
Our investigation commenced with the evaluation of optimal reaction conditions for the formation of the key iminium triflate intermediate, **Int-A**, through the deoxygenative semi-reduction of secondary amide **1a** (see Supplementary Information, section 5.1)[48]. We found that subsequent addition of 2-fluoropyridine (1.2 equiv.), triflic anhydride (1.1 equiv.), and triethyl silane (1.1 equiv.) to a dichloromethane solution of amide **1a** (0.25 M), cooled in an ice bath, led to the quantitative yield of the corresponding iminium ion **Int-A** (see Supplementary Information, section 5.1).

Next, we turned our attention to developing a one-pot protocol to couple the deoxygenative event with the photochemical alkylative step. To the solution obtained from the activation step, cyclohexyl iodide (3.0 equiv.), tris(trimethylsilyl)silane (TTMS, 2.0 equiv.), and dichloromethane (0.05 M) were directly added. The resulting reaction mixture was irradiated with 390 nm light for 12 h, yielding the desired substituted secondary amine **4** in good yield (75%, Table 1, Entry 1). Shortening the reduction time during the activation step resulted in similar yield of **4** (76%, Table 1, Entry 2). Notably, lower yields were observed when either visible light (456 nm) or azobisisobutyronitrile (AIBN) was used to promote the radical chain alkylative event (Table 1, Entries 3–4)[56]. Additional screening revealed that protic additives such as hexafluoro-2-propanol (HFIP) and *iso*-propanol negatively impacted the reaction outcome, resulting in a considerable amount of the fully reduced unbranched amine (Table 1, Entry 5, see Supplementary Table 6). Further refinement of the reaction conditions showed that using a dichloromethane-acetonitrile solvent mixture (1:4, v:v) resulted in higher yields (80%, Table 1, Entry 6), while ethyl acetate afforded lower results (Table 1, Entry 7). Control experiments performed in the dark or without TTMS resulted in the quantitative recovery of iminium triflate **Int-A**, respectively (Table 1, Entries 8–9).

### Substrate scope
With optimal conditions in hand, we first evaluated the influence of the amine fragment on the generality of the deoxygenative alkylation of secondary benzamides using isopropyl iodide (*i*Pr–I) as the coupling partner (Fig. 2). A variety of secondary *N*-alkyl groups, featuring functional groups such as ethers, esters and protected amines, performed well in the one-pot protocol, affording the desired α-substituted secondary amines in good yields (**7**–**11**). Similarly,

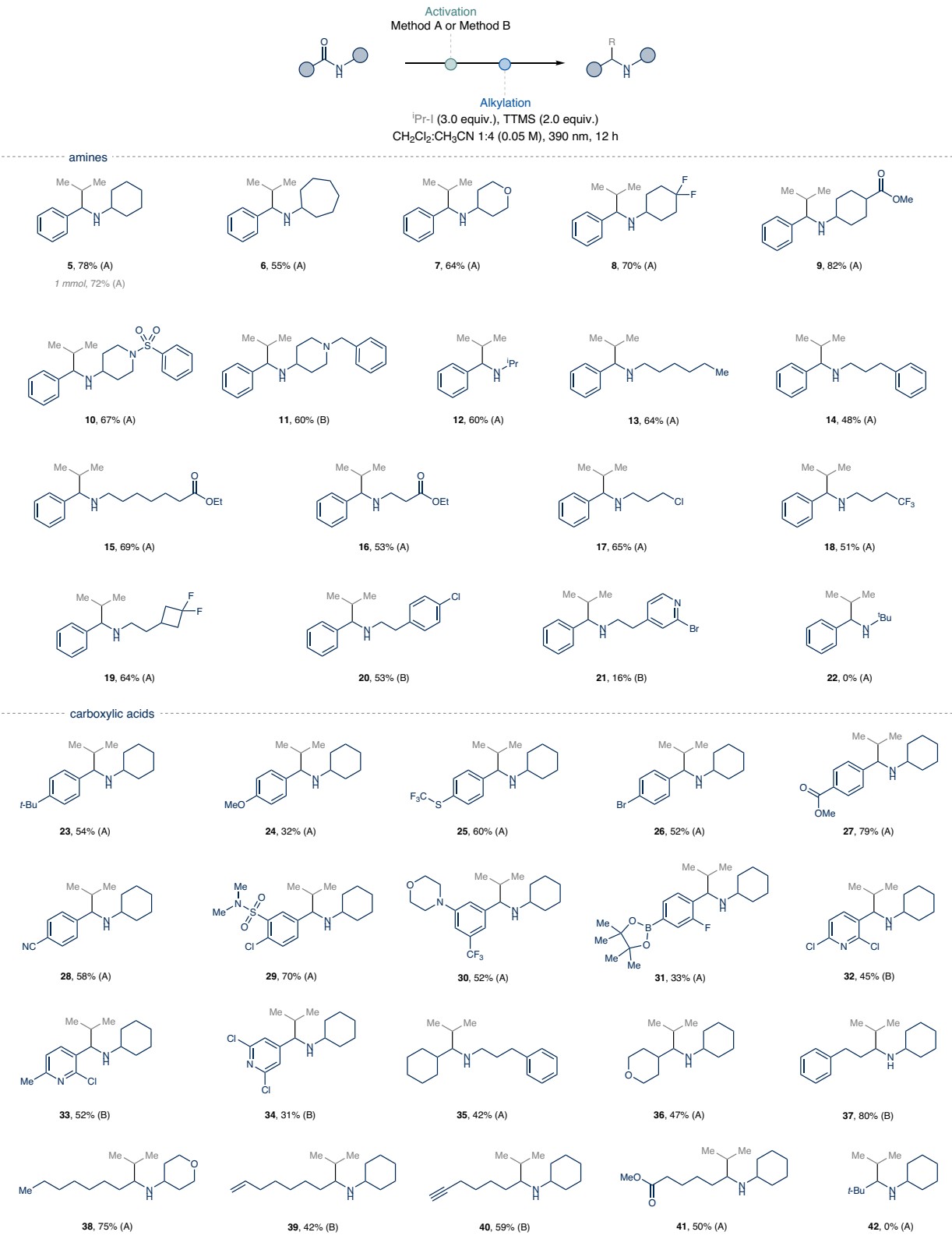

**Fig. 2 | Scope of the deoxygenative photochemical alkylation of secondary amides varying either the amine or the carboxylic acid fragments.** Method A: sec-amides (0.25 mmol, 1.0 equiv.), 2-FPy (1.2 equiv.), Tf₂O (1.1 equiv.), CH₂Cl₂ (0.25 M), 0 °C, 20 min then Et₃SiH (1.1 equiv.), 0 °C to rt, 5 h. Method B: sec-amides (0.25 mmol, 1.0 equiv.), 2-FPy (1.2 equiv.), Tf₂O (1.1 equiv.), CH₂Cl₂ (0.25 M), −78 °C to 0 °C, 20 min then Et₃SiH (1.1 equiv.), 0 °C to rt, 5 h. For the alkylation step and further experimental details see the Supplementary Information.

**Fig. 3 | Scope of the deoxygenative photochemical alkylation of sec-amides varying the alkyl iodides.** Activation: sec-amides (0.25 mmol, 1.0 equiv.), 2-FPy (1.2 equiv.), Tf$_2$O (1.1 equiv.), CH$_3$CN (0.25 M), 0 °C, 20 min then Et$_3$SiH (1.1 equiv.), 0 °C to rt, 5 h. For further experimental details see the Supplementary Information.

primary N-alkyl secondary amides were efficiently converted to the corresponding products in moderate to good yields (**13–15**). Notably, primary amines with vicinal electron-withdrawing groups, such as esters (**16**) and trifluoromethyl groups (**18**), showed clean reaction profiles. The mild conditions of the protocol tolerated the presence of chlorine atoms (**17**), providing a useful synthetic handle for further synthetic diversification. For N-phenyl ethyl units, performing the activation step at cryogenic temperature (−78 °C) was necessary to prevent Bischler–Napieralski-type side reactions[57], leading to the corresponding α-substituted phenylethylamines (**20–21**).

We next examined the deoxygenative functionalization of N-cyclohexyl amides, varying the aromatic carboxylic acid partner (Fig. 2, **23–34**). A wide array of commercially available benzoic acids with different electronic properties performed well under our optimized conditions. Remarkably, substrates decorated with functional groups such as ester (**27**), nitrile (**28**), sulfonamide (**29**), tertiary amine (**30**), halogens (**26 and 29**), and boronic ester (**31**) were also competent reaction partners. The process was also amenable for accessing halogenated N-heterobenzylamines in moderate yields (**32–34**). Collectively, this technology offers an intuitive and facile strategy to synthesize the medicinally relevant benzylamine cores with α–substitution (**23–34**)[11,58].

Moreover, we were pleased to find that N-cyclohexyl aliphatic amides were effective substrates, affording fully aliphatic α-substituted secondary amines in moderate to good yields (Fig. 2, **35–41**). Importantly, the deoxygenative reaction tolerated radical-sensitive unsaturated groups, such as alkenes and alkynes (**39–40**), providing useful synthons for building molecular complexity. Notably, sterically hindered amides, on either the amine or carboxylic acid moieties, remained unreactive under the optimized conditions, leading to the quantitative recovery of the iminium triflate (**22, 42**).

Then, N-cyclohexyl benzamide **1a** was used as model substrate to explore the scope of alkyl iodides (Fig. 3). A variety of primary and secondary alkyl iodides, including small rings and O, S, N-heterocycles, were successfully employed, delivering the corresponding α-alkylated secondary amines (**43–52**). This protocol further enables the efficient introduction of strained tertiary alkanes like bicyclo[1.1.1]pentanes (BCPs), potentially establishing a new strategy to access diphenylmethyl amine bioisosteres (**53–54**)[59].

Having demonstrated the generality and high functional group tolerance of this photochemical deoxygenative alkylation method, we next investigated its application to the late-stage modification of drug-like amides (Fig. 4A)[60–62]. Derivatives of ataluren and related oxadiazole analogues were successfully converted to the corresponding α-alkylated amines (**55, 56, 59**). Additionally, medicinally relevant N-containing heterocycles such as triazole, pyrazoles, and pyrimidine, as well as marketed drugs including adapalene and probenecid acid, were functionalized under optimized reaction conditions, yielding complex secondary amines in good yields (**57, 58–63**).

Finally, we showcase the modularity of the protocol through the synthesis of α-branched N-substituted cyclic tertiary amines (Fig. 4B, **64–71**). Specifically, inexpensive bis-electrophiles were used for the preparation of secondary amides bearing a pendant leaving group. This approach facilitates the rapid access to decorated pyrrolidines and piperidines via a formal deoxy-alkylative annulation strategy[63,64]. The saturated N-heterocycles were obtained without prior chromatographic purifications of the alkylated secondary amines. Spontaneous cyclization afforded pyrrolidine derivatives (**64–67**) upon basic aqueous work-up. In contrast, the 6-exo annulation was triggered by heating the acetonitrile solution containing the amine precursor, NaHCO$_3$, and sodium iodide (**68–71**).

## Mechanistic insights and scale-up

In order to gain mechanistic insights into the developed two-step, one-pot deoxygenative photochemical alkylation of secondary amides, we conducted a series of additional experiments (Fig. 5A). The presence of free-radical species was confirmed by a radical trapping experiment using 2,2,6,6–tetramethylpiperidine-1-oxyl (TEMPO): while adduct **72** was detected via GC-MS, the formation of the desired product **5** was completely inhibited[65]. In a radical clock experiment, subjecting amide **73** to standard reaction conditions resulted in the formation of pyrrolidine **75** as the only detected product. This suggests the initial formation of an aminium radical cation capable of undergoing 5-exo-trig cyclization with the pendant olefin[26,66–68]. Taken together, and based on literature insights[13,69,70], these results suggest that a light-promoted radical addition to iminium triflate **76** yields an electrophilic aminium radical cation **77**, which undergoes a polarity matched hydrogen atom transfer with TTMS to afford the ammonium product **78**. The formed silyl radical is postulated to perform a halogen atom transfer (XAT)

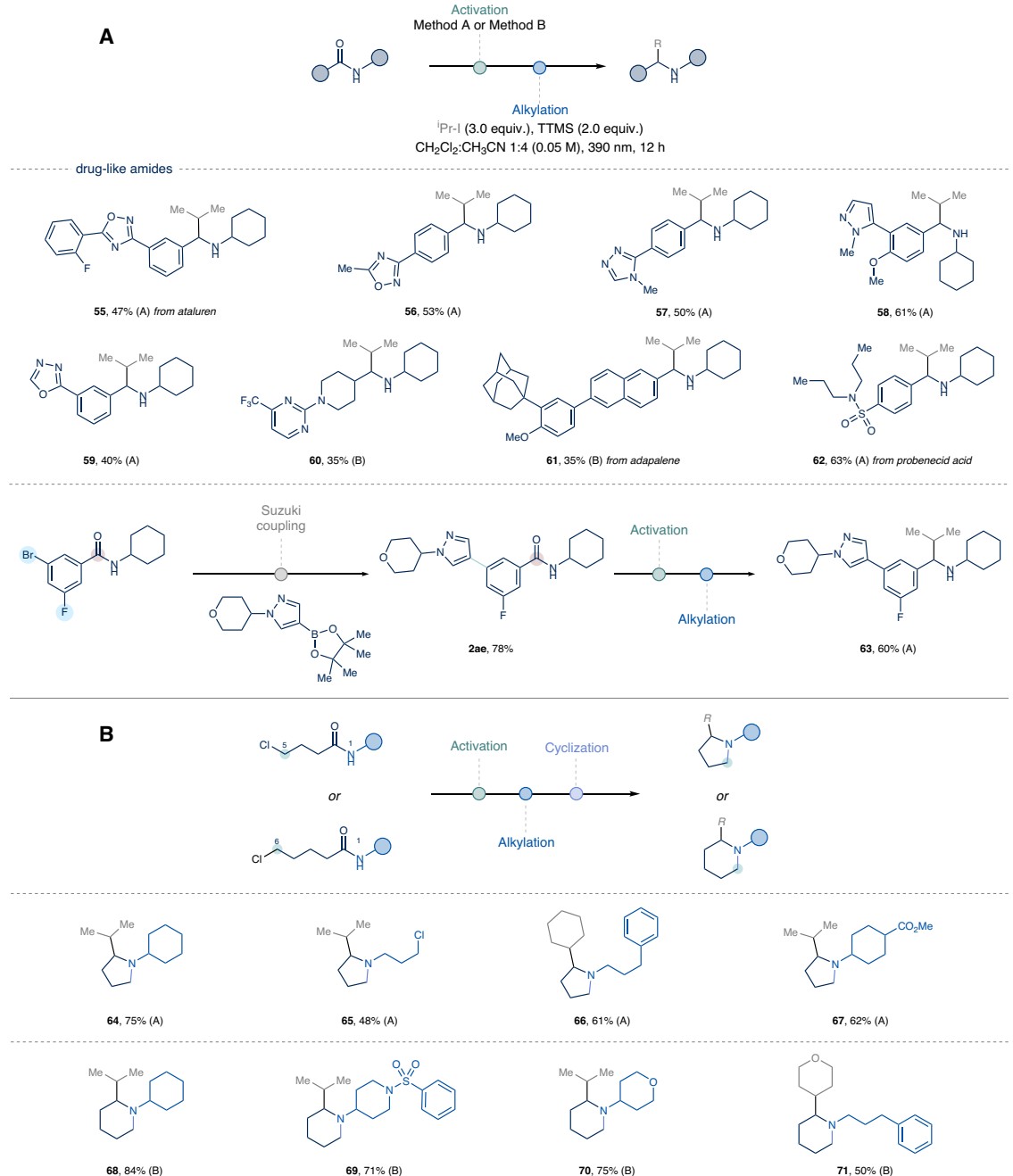

**Fig. 4 | Late-stage functionalization and alkylative annulation strategy.**
**A** Deoxygenative photochemical alkylation of drug-like sec-amides. **B** Deoxy-alkylative annulation strategy for the synthesis of α-branched N-substituted cyclic tertiary amines. Method A: sec-amides (0.25 mmol, 1.0 equiv.), 2-FPy (1.2 equiv.), Tf$_2$O (1.1 equiv.), CH$_2$Cl$_2$ (0.25 M), 0 °C, 20 min then Et$_3$SiH (1.1 equiv.), 0 °C to rt, 5 h. Method B: sec-amides (0.25 mmol, 1.0 equiv.), 2-FPy (1.2 equiv.), Tf$_2$O (1.1 equiv.), CH$_2$Cl$_2$ (0.25 M), −78 °C to 0 °C, 20 min then Et$_3$SiH (1.1 equiv.), 0 °C to rt, 5 h. For the alkylation step and further experimental details see the Supplementary Information.

step on isopropyl iodide, thus sustaining the radical chain process (Fig. 5B).

Given the synthetic relevance of this protocol and our interest to bridge the gap between academic discovery and industrial implementation through technology, we re-optimized the reaction conditions to achieve a more sustainable and efficient process (Fig. 5C). Major process chemistry pitfalls were identified in the use of dichloromethane as a solvent in the activation phase, long reaction times under light irradiation, and the stoichiometry of the binary alkylating system (TTMS and alkyl iodide)[71]. After re-optimization of these key parameters, we observed that the transformation could be scaled up to 5 mmol in batch, albeit with a slightly reduced yield (**5**, 54%). Specifically, the telescoped process proceeded at higher concentration (0.25 M) using acetonitrile as the only solvent and employed stoichiometric quantities of the alkylating agents. To address the sluggish reaction times and reduced yield of the photochemical step, we first investigated the impact of light intensity. Results indicated that higher photon fluxes correlated with increased reaction rates (See Supplementary Table 13)[72]. Based on these observations, we hypothesized that translating the photochemical alkylative step to continuous flow would grant improved efficiency and scalability[73,74]. Pleasingly, after brief optimization of the reaction

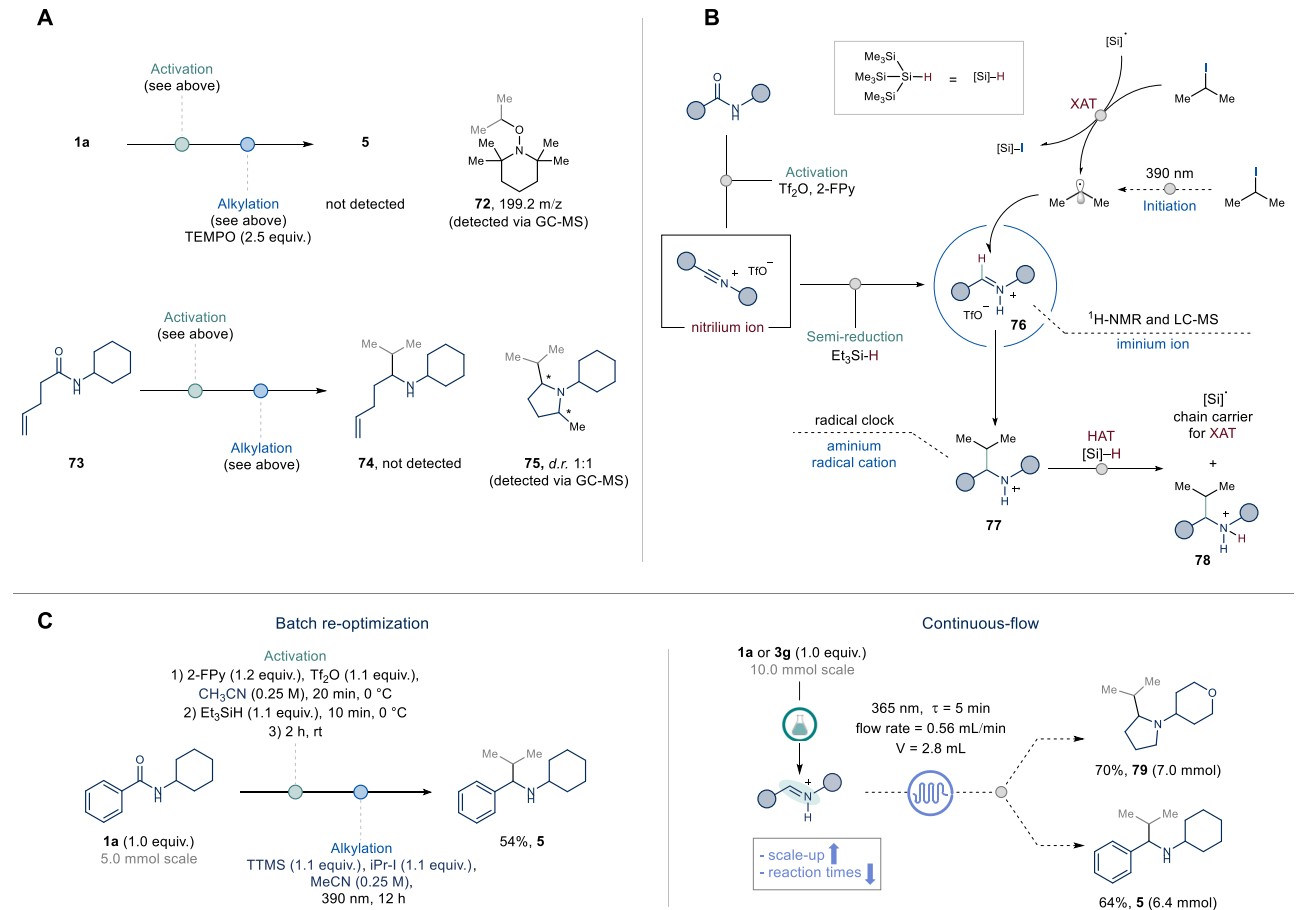

**Fig. 5 | Mechanistic investigation and scale-up in flow. A** Radical clock experiments. **B** Proposed mechanism. **C** Streamlined process enabling scale-up.

parameters (See Supplementary information, section 10.2), the model α-branched amine **5** was obtained in good isolated yield at 10 mmol scale (64% yield). The robustness and generality of the flow protocol were further demonstrated by the gram-scale synthesis of pyrrolidine **79** (70% yield), obtained after direct basic aqueous work-up of the reaction effluent (See Supplementary information, section 10.2.3).

## Discussion

In conclusion, we have developed a robust and versatile deoxygenative photochemical alkylation method for the efficient synthesis of α-substituted secondary amines from secondary amides. This innovative approach demonstrates broad functional group tolerance and has been successfully applied to the late-stage modification of drug-like molecules, offering significant advantages over traditional aldehyde-based methodologies. Our mechanistic studies confirm the involvement of radical intermediates, and process optimization has enabled scalability and enhanced efficiency, particularly through continuous-flow technology. This protocol not only expands the synthetic toolbox for medicinal chemistry but also bridges the gap between academic discovery and industrial application, paving the way for streamlined and sustainable production of complex amine derivatives.

## Methods

### General procedure for the deoxygenative alkylation of amides (activation at 0 °C)

*Step 1.* In a typical experiment, an oven-dried 7 mL vial equipped with a stirring bar was added the corresponding amide (0.25 mmol, 1.0 equiv.), and the vial was sealed with a septum. Subsequently, dry and degassed $CH_2Cl_2$ (1.0 mL) was added under $N_2$ atmosphere (0.25 M). The mixture was cooled at 0 °C with an ice-water bath and 2-fluoropyridine (29 mg, 26 µl, 0.30 mmol, 1.2 equiv.) was added. Triflic anhydride (78 mg, 47 µl, 0.28 mmol, 1.1 equiv.) was added slowly dropwise and the mixture was stirred (900 rpm) for 20 min at 0 °C. Triethylsilane (32 mg, 44 µl, 0.28 mmol, 1.1 equiv.) was added dropwise and the resulting mixture was stirred at 0 °C for additional 10 min. Then, the vial was removed from the ice-water bath and left stirring at room temperature for 2–5 h (the iminium ion formation can be monitored via UPLC-MS).

*Step 2.* After, the solution was diluted with acetonitrile (4 mL, 0.05 M *final concentration*) and the corresponding alkyl iodide (0.75 mmol, 3.0 equiv.) and tris(trimethylsilyl)silane (124 mg, 154 µl, 0.50 mmol, 2.0 equiv.) were added. The vial was sealed with electrical tape and stirred and irradiated under 390 nm in the UFO photochemical reactor for 12 h. The temperature was maintained at 30–35 °C during the course of the reaction.

*Step 3.* Finally, the vial was removed from the photochemical reactor and the solvent was evaporated under reduced pressure. The crude was suspended in *n*-pentane (10 mL) and sonicated for 2 min. The surnatant was filtered through a plug of celite to retain solid traces. This process was repeated three times. Finally the celite plug was washed with $CH_2Cl_2$ (3 × 5 mL). The $CH_2Cl_2$ phases were collected and added to the residual solid. The corresponding solution was then diluted with 15 mL of $NaHCO_3$ (~1:1 ratio organic: water phase) and the biphasic mixture was stirred for 15 min at rt. The solution was then transferred to a separatory funnel and extracted with $CH_2Cl_2$ (3 × 15 mL). The combined organic layers were dried over $Na_2SO_4$, filtered and the solvent was removed under reduced pressure.

The crude reaction mixture was purified by flash column chromatography on silica gel.

## General procedure for the deoxygenative alkylation of amides and subsequent cyclization for the synthesis of *N*-substituted piperidines

*Alkylation*: In a typical experiment, to an oven-dried 7 mL vial equipped with a stirring bar was added the corresponding amide (0.25 mmol, 1.0 equiv.), and the vial was sealed with a septum. Subsequently, dry and degassed $CH_2Cl_2$ (1.0 mL) was added under $N_2$ atmosphere (0.25 M). The mixture was cooled at 0 °C with an ice-water bath and 2-fluoropyridine (29 mg, 26 μl, 0.30 mmol, 1.2 equiv.) was added. Triflic anhydride (78 mg, 47 μl, 0.28 mmol, 1.1 equiv.) was added slowly dropwise and the mixture was stirred (900 rpm) for 20 min at 0 °C. Triethylsilane (32 mg, 44 μl, 0.28 mmol, 1.1 equiv.) was added dropwise and the resulting mixture was stirred at 0 °C for additional 10 min. Then, the vial was removed from the ice-water bath and left stirring at room temperature for 5 h. After, the solution was diluted with acetonitrile (4 mL, 0.05 M *final concentration*) and the corresponding alkyl iodide (0.75 mmol, 3.0 equiv.) and tris(trimethylsilyl)silane (124 mg, 154 μl, 0.50 mmol, 2.0 equiv.) were added. The vial was sealed with electrical tape and stirred and irradiated under 390 nm in the UFO photochemical reactor for 12 h. The temperature was maintained at 30–35 °C during the course of the reaction. Finally, the vial was removed from the photochemical reactor and the solvent was evaporated under reduced pressure. The crude was suspended in *n*-pentane (10 mL) and sonicated for 2 min. The surnatant was filtered through a plug of celite to retain solid traces. This process was repeated three times. Finally, the celite plug was washed with $CH_2Cl_2$ (3 × 5 mL). The $CH_2Cl_2$ phases were collected and added to the residual solid. The corresponding solution was then diluted with 15 mL of $NaHCO_3$ (-1:1 ratio organic: water phase) and the biphasic mixture was stirred for 15 min at rt. The solution was then transferred to a separatory funnel and extracted with $CH_2Cl_2$ (3 × 15 mL). The combined organic layers were dried over $Na_2SO_4$, filtered and the solvent was removed under reduced pressure.

*Cyclization:* Then, anhydrous $NaHCO_3$ (42 mg, 0.50 mmol, 2.0 equiv.), NaI (75 mg, 0.125 mmol, 0.5 equiv.) and a stirring bar were added to the flask containing the crude reaction mixture and it was sealed with a rubber septum. Next, dry acetonitrile was added (2.5 mL, 0.1 M) and the flask was placed in an oil bath at 80 °C for 2 h. The reaction mixture was cooled to rt and filtered using a sintered funnel, rinsed with dichloromethane and the solvent was removed under reduced pressure. The crude reaction mixture was purified by flash column chromatography on silica gel.

*Generally, we observed a yield loss (5–10% lower than ¹H NMR yields) due to product isolation via column chromatography, particularly for tertiary amines (cyclization protocol, Fig. 4B).*

## Data availability

The data supporting the results of the article, including optimization studies, experimental procedures, compound characterization, mechanistic studies, and scale-up procedures are provided within the paper and its Supplementary Information. Additional data are available from the corresponding author upon request.

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

## Acknowledgements

We are grateful to have received generous funding from the European Union H2020 research and innovation program under the Marie S. Curie Grant Agreement (CHAIR, No 860762, A.P., M.J., T.N.; PhotoReAct, No 956324, S.B., D.D., T.N.). We also would like to thank Dr. Jesus Alcazar for stimulating discussions and Ed Zuidinga for help with the HRMS measurements.

## Author contributions

A.P. and S.B. designed the project, with input from T.N. A.P., S.B., R.S., and D.D. performed and analyzed the synthetic experiments with input from M.T.T., M.J., A.J.W. and T.N. A.P., S.B. and T.N. wrote the manuscript with input from all the authors.

## Competing interests

The authors declare no competing interests.
