## [Transparent Peer Review file · Nature Communications]

Deoxygenative Photochemical Alkylation of Secondary Amides Enables a Streamlined Synthesis of Substituted Amines

Corresponding Author: Professor Timothy Noel

Version 0:

Reviewer comments:

Reviewer #1

(Remarks to the Author)

In this paper Noel and co-workers report on a radical-based method for the conversion of amides via iminium species into secondary amines. This is achieved via two stages where an initial triflation of the amide and a semi-reduction with a silane is followed by a photochemical C-alkylation of the iminium species. Overall, this protocol gives access to a variety of secondary amines that are relevant building blocks for bioactive compounds. The substrate scope is wide showing tolerance of a variety of medically relevant functionalities. As the yields vary around 50-60%, it would be useful if the authors could discuss whether yield loss is due to side product formation or unreacted amide substrate etc.

The authors point out that alternative methods to access similar secondary amines are limited due to selectivity (i.e. N-alkylation) and operational complexity (i.e., reductive amination). While this is agreed for N-alkylations, one might argue that reductive aminations are attractive as they are reliable and the use of aldehydes and standard reductants (NaBH₄, NaCNBH₃, Pd/C + H₂ etc.) are robust methods that can be used on scale in industry. While amides are readily available from acids (that are more abundant and stable than many aldehydes) and amines, one should consider that the amide forming step and the chemical waste generated needs to be taken into account when comparing the pros and cons of such methods. In the presented methodology, there appears to be a need for various reagents and additives (some in superstoichiometric amounts) that amount to significant amounts of chemical waste, which is compounded by using iodides as radical precursors. This may be tolerable for small scale library synthesis, but the overall low atom efficiency may present a bottleneck.

It is noted that the authors demonstrate a re-optimization in Fig. 5 (c) which attempts to address this issue, however a yield drop of 20% is observed as a consequence.

The addition of a flow stage for the second part of the process is interesting, however, it is less clear why the authors have not developed a joint approach including the amide semi reduction step given that this would be more appealing.

As some minor items to consider, the role of 2-fluoropyridine could be clarified given that it appears to be an unusual pyridine variant. Also, in Fig. 1 (b) retrosynthetic arrows may be more appropriate than the dashed arrows to indicate where the encircled target can be derived from. Lastly, in the ESI files please check the spectra provided as in some case 1H-spectra are inadvertently given instead of 13C spectra (e.g. compound 10, page 114).

Overall, the presented methodology is sound and interesting from a synthetic standpoint and it appears to work well across a large set of substrates. This would enable its use for industrial library synthesis affording small amounts of valuable amine products. The manuscript is well described and accompanied by a detailed ESI file which should allow for simple reproduction of the work.

Reviewer #2

(Remarks to the Author)

This paper by Noël and coworkers builds on the well-established strategy in which carbon-based radical species add to iminium salts. This approach, initially demonstrated by Gaunt (refs 13, 26) and others, is effectively leveraged here. Noël's team capitalized on the robust methodology for iminium salt formation from amides (as developed by Gosez, Charette, Maulide, Movassaghi among others), followed by in situ reduction to imines (as shown by Charette, refs 44, 45) alongside

photo-mediated carbon radical generation. This innovative combination enables a streamlined synthesis of substituted amines from amides. Remarkably, the radical addition proceeds efficiently even in the presence of numerous by-products generated during the amide activation and reduction steps. Overall, this transformation offers considerable value to synthetic organic chemists due to the accessibility of amides as starting materials. I recommend this work for publication in Nature Communications.

Additional comments or questions:

- Were any 2-fluoropyridine addition products observed?
- The figure describing the flow experiment should include more details. It would be helpful for readers to clearly see that all components were pre-mixed in a single feed tank and pumped through the photo-flow reactor.

Reviewer #3

(Remarks to the Author)

Noël and coworkers have presented a robust and versatile method for the efficient transformation of secondary amide into α -branched secondary amines. This approach is a two-step strategy by combination of the reported triflic anhydride-mediated semi-reduction of amides to N-alkyl iminium ions and the silane-mediated halogen atom transfer facilitated photochemical alkyl radical addition to iminium ions with alkyl halides. The broad substrate scope, application on late-stage modifications of drug-like molecules and synthesis of saturated N-heterocycles, and scalability enabled by flow technology are impressive. However, this referee still has some concerns about this manuscript :

1. The referee noticed that Wang group reported a deoxygenative alkylation of tertiary amides via merging Ir catalyzed amides reduction with a visible light induced radical generation from alkyl iodides (Feng Zhao, Feng Jiang, Xiaoming Wang. *Sci. China Chem.* 2022, 65, 2231–2237.), which is not cited in this manuscript. This manuscript and that by Wang reported methods for generating iminium ion intermediate from secondary and tertiary amides, respectively, and the following key alkyl radical addition step facilitated by silane-mediated halogen atom transfer were the same. Given this precedent and the close relationship between the two, my enthusiasm was diminished.
2. For the substrate scope, several amides such as those derived from aryl acid and aryl amine or from linear alkyl acid with linear alkyl amine should be examined.
3. The following articles for deoxygenative alkylation of amides should be cited. (a) Xiyike Deng, Feng Jiang, Xiaoming Wang. *Org. Lett.* 2024, 26, 2483–2488. (b) Tatiana Rogova, Pablo Gabriel, Stamatia Zavitsanou, Jamie A. Leitch, Fernanda Duarte, Darren J. Dixon. *ACS Catal.* 2020, 10, 11438–11447. (c) Jiexiang Lu, Zhenghua Li, Li Deng. *J. Am. Chem. Soc.* 2024, 146, 4357–4362.
4. The references citation format is not consistent, which should follow the requirement of Nature Communication.

Version 1:

Reviewer comments:

Reviewer #1

(Remarks to the Author)

The authors have addressed all comments from this reviewer in their response letter. It would be recommended to also add comments in the revised manuscript itself pertaining to known reasons for lower isolated yields due to isolation issues or side reactions rather than 'just' acknowledging this to the reviewers.

In relation to the problematic atom economy when performing the amide semi-reduction step, the authors state that future efforts will use acids and amines to directly form the required imines. It is not clear how this will solve the issue given that bespoke reagents will still be needed to affect the selective reduction of the acid to an aldehyde (equivalent). Details should be added to not have this as speculative.

Reviewer #2

(Remarks to the Author)

The authors have addressed all my points. I recommend publication as is.

Reviewer #3

(Remarks to the Author)

Noël and coworkers have addressed all the concerns raised by the referees. This revised manuscript could be considered for publication in *Nat. Commun.* However, this referee still concerns about the scope of alkyl radical precursors. In this manuscript, the authors utilized alkyl iodides, which is generally less stable and more expensive than their chloride, or bromide counterpart. Comments on the use of alkyl bromides, or other type precursors could be valuable to the readers and

potential users.

Comments to the editor

The main text and references have been revised (and highlighted) in accordance with the formatting guidelines outlined by Nature Communications.

Reviewer 1

In this paper Noel and co-workers report on a radical-based method for the conversion of amides via iminium species into secondary amines. This is achieved via two stages where an initial triflation of the amide and a semi-reduction with a silane is followed by a photochemical C-alkylation of the iminium species. Overall, this protocol gives access to a variety of secondary amines that are relevant building blocks for bioactive compounds. The substrate scope is wide showing tolerance of a variety of medicinally relevant functionalities.

We kindly thank the Reviewer for these comments.

As the yields vary around 50-60%, it would be useful if the authors could discuss whether yield loss is due to side product formation or unreacted amide substrate etc.

The amide starting material is always fully converted after the thermal semi-reduction (Tf_2O and Et_3SiH), generally affording the targeted iminium ion in nearly quantitative yield. However, in some cases side-products from Bishler-Napieralski reaction or nitrile formation via extrusion of the N-substituent as carbocation were observed. To address this issue, we found that lowering the temperature to $-78\text{ }^\circ\text{C}$ can attenuate such side reactions.

We observed that a yield loss (5–10% lower than 1H NMR yields) is due to product isolation via column chromatography, especially for tertiary amines (cyclization protocol, Figure 4B).

As for the photochemical step, the only side-product that we could identify and isolate is the unbranched amine derived from reduction of the iminium ion. In this case, the side product can account for approximately 5% of the total mass balance.

The authors point out that alternative methods to access similar secondary amines are limited due to selectivity (i.e. N-alkylation) and operational complexity (i.e., reductive amination). While this is agreed for N-alkylations, one might argue that reductive aminations are attractive as they are reliable and the use of aldehydes and standard reductants ($NaBH_4$, $NaCNBH_3$, $Pd/C + H_2$ etc.) are robust methods that can be used on scale in industry. While amides are readily available from acids (that are more abundant and stable than many aldehydes) and amines, one should consider that the amide forming step and the chemical waste generated needs to be taken into account when comparing the pros and cons of such methods. In the presented methodology, there appears to be a need for various reagents and additives (some in superstoichiometric amounts) that amount to significant amounts of chemical waste, which is compounded by using iodides as radical precursors. This may be tolerable for small scale library synthesis, but the overall low atom efficiency may present a bottleneck.

Although reductive amination is a powerful and reliable synthetic strategy when using aldehydes to obtain unbranched secondary amines, the synthesis of α -branched analogues necessitates the use of unsymmetrical

ketones that are generally accessed by multistep synthetic sequences. Additionally, reductive amination of ketones features a more sluggish condensation step.

We agree with the reviewer on the fact that amide bond formation is still associated with considerable chemical waste (coupling agents), however our strategy offers a possibility to repurpose the extensive libraries of amides already present in pharmaceutical companies, ultimately yielding α -branched secondary amines in a more modular way.

In agreement with this comment from the Reviewer, future endeavors are focused on generating the desired iminium ions directly from carboxylic acids and amines, without prior synthesis and isolation of the amide. Furthermore, it is important to highlight that the scale-up was achieved upon re-optimization of the reaction conditions. The revised conditions enhance the overall efficiency of the system, reducing the amount of supersilane and isopropyl iodide from super superstoichiometric to stoichiometric level, increasing five times the final concentration and employing acetonitrile as the sole solvent during the entire reaction sequence.

It is noted that the authors demonstrate a re-optimization in Fig. 5 (c) which attempts to address this issue, however a yield drop of 20% is observed as a consequence. The addition of a flow stage for the second part of the process is interesting, however, it is less clear why the authors have not developed a joint approach including the amide semi reduction step given that this would be more appealing.

As for the 20% drop in yield after batch re-optimization, we noticed a positive impact of light intensity on the reaction yield (See Supplementary Table 13), thus we opted for the execution of the photochemical step in flow, ultimately proving higher yielding (64% isolated, 70% ^1H NMR yield).

We observed that the thermal semi reduction step was scalable and reproducible in batch across all the tested scales (0.25-10 mmol), while its translation to continuous flow presented some problematic aspects: the preparation of stock solutions of triflic anhydride proved challenging as Tf_2O reacted exothermically with the solvent upon standing (acetonitrile).

As some minor items to consider, the role of 2-fluoropyridine could be clarified given that it appears to be an unusual pyridine variant.

As reported by Charette (see refs 48 and 49), 2-fluoropyridine is generally the best performing across the series due to its low nucleophilicity and basicity. However, as reported in literature, other bases such as 2-chloro pyridine, 2-bromo pyridine and 2,6 dichloropyridine can be employed obtaining similar results.

Also, in Fig. 1 (b) retrosynthetic arrows may be more appropriate than the dashed arrows to indicate where the encircled target can be derived from.

We changed the arrows as requested.

Lastly, in the ESI files please check the spectra provided as in some case ^1H -spectra are inadvertently given instead of ^{13}C spectra (e.g. compound 10, page 114).

We are grateful for this observation and made the required corrections.

Reviewer 2

This paper by Noël and coworkers builds on the well-established strategy in which carbon-based radical species add to iminium salts. This approach, initially demonstrated by Gaunt (refs 13, 26) and others, is effectively leveraged here. Noël's team capitalized on the robust methodology for iminium salt formation from amides (as developed by Gosez, Charette, Maulide, Movassaghi among others), followed by in situ reduction to imines (as shown by Charette, refs 44, 45) alongside photo-mediated carbon radical generation. This innovative combination enables a streamlined synthesis of substituted amines from amides. Remarkably, the radical addition proceeds efficiently even in the presence of numerous by-products generated during the amide activation and reduction steps. Overall, this transformation offers considerable value to synthetic organic chemists due to the accessibility of amides as starting materials. I recommend this work for publication in Nature Communications.

We kindly thank the Reviewer for these comments.

Additional comments or questions:

- Were any 2-fluoropyridine addition products observed?

We did not detect any product of radical addition to 2-fluoropyridine via Minisci-type reaction. Specifically, after the semi-reduction step, the solution contains the iminium triflate and the non-protonated 2-fluoropyridine. For this reason, radical addition occurs on the iminium ion and not on the 2-F-pyridine. Additionally, the absence of an external oxidant precludes oxidative re-aromatization in a Minisci-type reactivity to yield the alkylated 2-fluoropyridine.

- The figure describing the flow experiment should include more details. It would be helpful for readers to clearly see that all components were pre-mixed in a single feed tank and pumped through the photo-flow reactor

We thank the reviewer for this suggestion. We have modified the scheme accordingly.

Reviewer 3

Noël and coworkers have presented a robust and versatile method for the efficient transformation of secondary amide into α -branched secondary amines. This approach is a two-step strategy by combination of the reported triflic anhydride-mediated semi-reduction of amides to N-alkyl iminium ions and the silane-mediated halogen atom transfer facilitated photochemical alkyl radical addition to iminium ions with alkyl halides. The broad substrate scope, application on late-stage modifications of drug-like molecules and synthesis of saturated N-heterocycles, and scalability enabled by flow technology are impressive.

We kindly thank the Reviewer for these comments.

However, this referee still has some concerns about this manuscript :

1. The referee noticed that Wang group reported a deoxygenative alkylation of tertiary amides via merging Ir catalyzed amides reduction with a visible light induced radical generation from alkyl iodides (Feng Zhao, Feng Jiang, Xiaoming Wang. *Sci. China Chem.* 2022, 65, 2231–2237.), which is not cited in this manuscript. This manuscript and that by Wang reported methods for generating iminium ion intermediate from secondary and tertiary amides, respectively, and the following key alkyl radical addition step facilitated by silane-mediated halogen atom transfer were the same. Given this precedent and the close relationship between the two, my enthusiasm was diminished.

*We thank the referee for pointing out this reference and we have added it in the main manuscript (ref. 43). However, while several strategies based on the radical Mannich type addition on alkyl iminium ions were developed for the synthesis of α -branched tertiary amines, the analogous version for the preparation of α -branched secondary amines are underrepresented (see ref. 16, 28 from Gaunt group) and during the development of our method no radical version was known. Accordingly, we considered secondary amides as privileged precursors for their introduction. The Wang group exploited the Vaska complex as an efficient catalyst for the semi-reduction of the parent tertiary amide. This complex is highly selective for tertiary amides, usually delivering quantitative formation of the corresponding enamine intermediates. We want to stress that the Vaska complex presents sluggish reactivities towards secondary amides as reported by the Dixon group (see ref. 45), thus representing a scope limitation in Wang's work. Few Iridium-based catalysts such as $[\text{Ir}(\text{COE})_2\text{Cl}]_2$ were reported by the Huang group as alternative complex for the semi-reduction of secondary amides (see ref. *Angew. Chem. Int. Ed.* 2018, 57, 11354–11358). However, the reported sensitivity to air of the complex represents a practical limitation (see ref. 45 and C. V. S. J. L. Herde. *J. C. Lambert, Inorganic Syntheses* 1974, 15, 18-20). Thus, we envision that a triflic anhydride mediated activation of sec-amides could offer a more robust platform for the generation of α -branched secondary amines.*

2. For the substrate scope, several amides such as those derived from aryl acid and aryl amine or from linear alkyl acid with linear alkyl amine should be examined.

*We thank the reviewer for these suggestions. The semi-reduction of amides derived from anilines proved unsuccessful under the reported reaction conditions and remain a limitation of the protocol. We have added the case study of N-phenyl benzamide as limitation of the scope in the supplementary information. This outcome can be rationalized based on mechanistic evidence reported by the group of Movassaghi (*J. Org. Chem.* 2009, 74, 1341–1344). In this seminal study, the authors demonstrated how benzamides derived from*

anilines (N-Aryl benzamides) did not present an IR signal at 2370 cm⁻¹ diagnostic of a nitrilium ion while displaying only an absorption band at 1600 cm⁻¹ typical of amidinium ions. Taken together, these results suggest that the amidine intermediate might not be reduced by triethyl silane to afford the targeted iminium ion.

The reported protocol was applied to amides derived from linear aliphatic carboxylic acids and linear primary amines in the context of the synthesis of tertiary amines. The protocol proved general with respect to chain length and when varying the alkyl iodides and we believe that is general also across different acids and amines.

Linear-Linear amides

3. The following articles for deoxygenative alkylation of amides should be cited. (a) Xiyike Deng, Feng Jiang, Xiaoming Wang. *Org. Lett.* 2024, 26, 2483–2488. (b) Tatiana Rogova, Pablo Gabriel, Stamatia Zavitsanou,

Jamie A. Leitch, Fernanda Duarte, Darren J. Dixon. ACS Catal. 2020, 10, 11438–11447. (c) Jiaxiang Lu, Zhenghua Li, Li Deng. J. Am. Chem. Soc. 2024, 146, 4357–4362.

We agree with the Reviewer. The suggested citations were implemented in the main manuscript (see new Refs. 42, 44, 45).

4. The references citation format is not consistent, which should follow the requirement of Nature Communication.

Indeed, we have changed it accordingly.

Reviewer 1

The authors have addressed all comments from this reviewer in their response letter. It would be recommended to also add comments in the revised manuscript itself pertaining to known reasons for lower isolated yields due to isolation issues or side reactions rather than 'just' acknowledging this to the reviewers.

We have added in the main manuscript (methods section) the reason for lower isolated yields compared to the assay yield (¹H NMR yield of the crude mixture).

In relation to the problematic atom economy when performing the amide semi-reduction step, the authors state that future efforts will use acids and amines to directly form the required imines. It is not clear how this will solve the issue given that bespoke reagents will still be needed to affect the selective reduction of the acid to and aldehyde (equivalent). Details should be added to not have this as speculative.

We considered the use of a different silane as a dual-role reagent, promoting both the amide coupling and the subsequent reduction of the iminium triflate to iminium ion. These details are not speculative, as further research is ongoing to enhance the overall efficiency of the method, with results to be reported in due course. Furthermore, as noted in the first round of the comments regarding atom efficiency: The revised conditions enhance the overall efficiency of the system, reducing the amount of supersilane and isopropyl iodide from super superstoichiometric to stoichiometric level, increasing five times the final concentration and employing acetonitrile as the sole solvent during the all sequence.

Reviewer 2

The authors have addressed all my points. I recommend publication as is.

We kindly thank the Reviewer.

Reviewer 3

Noël and coworkers have addressed all the concerns raised by the referees. This revised manuscript could be considered for publication in Nat. Commun. However, this referee still concerns about the scope of alkyl radical precursors. In this manuscript, the authors utilized alkyl iodines, which is generally less stable and more expensive than their chloride, or bromide counterpart. Comments on the use of alkyl bromines, or other type precursors could be valuable to the readers and potential users.

*The use of alkyl bromides as radical precursors was addressed in the Supplementary Information, Section 5.3, Supplementary Table 7, "Alkylation using alkyl bromides", reporting the optimization for the deoxygenative alkylation of the model amide **1a** using cyclohexyl bromide as alkylating agent. Pleasingly, under optimized conditions the desired α -branched secondary amine **4** was obtained in 67% ^1H NMR yield when using 3.0 equiv. of NaI and 3.0 equiv. of cyclohexyl bromide. Additionally, when carrying out the same transformation with only 1.1 equiv. of TTMS and 1.1 equiv. of cyclohexyl bromide (0.05 M in dichloromethane) without any external source of iodide (NaI), the targeted amine **4** was obtained in 41% yield.*